# How Well Does Australian Animal Welfare Policy Reflect Scientific Evidence: A Case Study Approach Based on Lamb Marking

**DOI:** 10.3390/ani13081358

**Published:** 2023-04-15

**Authors:** Charlotte H. Johnston, Vicki L. Richardson, Alexandra L. Whittaker

**Affiliations:** 1School of Biomedicine, Faculty of Health and Medical Sciences, University of Adelaide, Adelaide, SA 5000, Australia; charlotte.johnston@adelaide.edu.au; 2School of Animal and Veterinary Sciences, The University of Adelaide, Roseworthy, SA 5371, Australia

**Keywords:** animal welfare legislation, tail docking, castration, mulesing, sheep, Australia

## Abstract

**Simple Summary:**

Animal welfare policy regarding husbandry practices in sheep in Australia differs between states and territories. This dis-uniformity of the legislature can be confusing and limit the application of the law, particularly with growing pressure from the local and global community to improve animal welfare. The influence of scientific evidence contributing to the development of these policies is unclear. This article explores the Australian animal welfare legislature and the scientific evidence informing husbandry practices commonly performed at lamb marking.

**Abstract:**

The development and substance of animal welfare policy is subject to a range of social, cultural, economic, and scientific influences that commonly vary within and between countries. Discrepancies in policy can create confusion and mistrust among stakeholders and consumers and limit the ability to create a uniform minimum level of requirements to safeguard animal welfare, as well as create a level ‘playing field’ for farmers when trading with other jurisdictions. The livestock sector is receiving growing scrutiny globally for real and perceived violations of animal welfare, for example, the practice of mulesing in Australia. This article explores animal welfare legislation within Australia and how it reflects the scientific evidence surrounding routine husbandry practices in sheep, including tail docking, castration, and mulesing. While there is some variation between state and territory legislation, the most notable concern is the lack of enforceable recommendations surrounding the evidence-based use of analgesia and anaesthesia for painful husbandry procedures. The age at which these procedures are recommended to be performed is relatively consistent across Australian jurisdictions, but there is a marked difference compared to international legislation. The global context of animal welfare legislation, public perception, and producer perception of these procedures are also discussed, highlighting the difficulty of creating robust animal welfare legislation that promotes a good standard of welfare that is respected worldwide whilst being practical in an Australian setting given our unique geography and climatic conditions.

## 1. Introduction

The farming of animals, once widely accepted by society, is now under growing scrutiny as animal production becomes more intensive and social attitudes toward the use of animals changes. This scrutiny comes from numerous groups of differing social, scientific, and political backgrounds. These groups include animal rights organisations, farming stakeholders, companies reliant on trade markets, the public, animal welfare scientists, veterinarians and animal health professionals, and politicians representing local, state, or federal interests [1]. The influence of these groups on the overall welfare of the individual animal can be substantial and may be beneficial or detrimental. Balancing the interests of these groups and the welfare of the relevant animals can be delicate and fraught with conflict. Legal frameworks for animal welfare protection should provide the scaffolding upon which a consensus can be reached. This consensus should optimise animal welfare based on available scientific evidence and meet socially respected requirements to promote a high standard of production and welfare and maintain a social licence to operate. The reality is often somewhat different and can depend largely on prevailing economic and political interests at the time. It is worth noting that the very existence of an animal welfare legislative framework supposes a utilitarian approach to the use of animals in society [2]. Animal use for human benefit is allowed, provided there are conditions in place to minimise suffering and promote welfare. However, it is the balance between what level of suffering is ‘reasonable’ or ‘necessary’ that may be contested, and factors that play into this equation are not purely based on animal outcomes but may be human-centric such as practicality and economic feasibility [2,3].

Notwithstanding the need for policymakers to balance multiple, often competing interests, it is generally considered (and stated) that legal provisions have a basis in evidence. In considering issues of animal welfare, it would be assumed that this evidence is derived from animal welfare science [4]. However, there has been little examination of the extent to which Australian legislatures incorporate animal welfare science into policy, the extent of the uniformity of this incorporation across Australian jurisdictions, and how this contrasts with international policy. The latter has become of greater importance of late with the creation of trade agreements, for example, the Australia–United Kingdom Free Trade Agreement, where, in the animal welfare context, a country’s treatment of their animals may be highlighted on the world stage [5]. The recent signing of this agreement caused controversy in the UK due to the perceived lower standards of animal welfare in Australia owing to the continued practice of mulesing in Australia. In addition, several surveys in Australia and internationally have highlighted inconsistencies between consumer and farmer perceptions of mulesing [6] and other husbandry procedures and the relevant legislation [6,7,8,9]. A similar concern may arise with one of the world’s biggest trading blocs, the European Union (EU), in the forthcoming trade agreement [10].

In this article, we tackle the question of the linkage between science and animal welfare policy using a case study approach based on the practice of lamb marking. We do this by examining the available science around specific aspects of lamb marking to understand the extent of the weight of this evidence. The subordinate legislation related to this practice across the states and territories is then sourced for relevant provisions and to assess uniformity across these jurisdictions. Policy from selected international jurisdictions is also examined as a comparator. We then discuss the extent of the linkages of policy principles identified with the established science. We also explore the perceptions of farmers surrounding these procedures, as this is a driving influence on compliance with any legislative changes or evidence-based recommendations on husbandry procedures. Finally, we briefly discuss the challenges associated with assimilating science into policy and how this might practically be achieved. We conclude with the extent to which the Australian system appears to have achieved this in the case of lamb marking.

## 2. What Is Lamb Marking?

Marking is the common term used to describe practices to identify young stock and perform early procedures aimed at maintaining flock health and productivity. The procedures performed vary depending on the market, management style, tradition, culture, and the environment. In Australian systems, lambs are typically ear tagged with property ID and vaccinated against a variety of infectious diseases depending on the management system. Surgical procedures may also be performed at these times according to the production style and farmer preference. These procedures most commonly include tail docking, with or without mulesing, and castration in males not intended for breeding. Tail docking is the amputation of part of the tail and can be performed by applying a tight rubber ring to the tail, which induces ischaemic necrosis and sloughing of the tail; cutting the tail at the desired length between the vertebrae; or cutting the tail with a heated sharp knife to cauterise the wound after incision. Mulesing is the removal of skin around the perineum and tail. This technique is only performed in Australia and most commonly involves using a sharp knife or mulesing shears [11]. Castration is the removal of the testicles by either applying a tight rubber ring to the neck of the scrotum or using a clean, sharp knife to incise the scrotum and remove the testicles. Immunocastration is a newer alternative to traditional methods of castration that does not cause pain [12,13]. This method blocks the normal functioning of the hypothalamic–pituitary–gonadal axis by administering two doses of vaccine against gonadotrophin-releasing hormone (GnRH). Immunisation against GnRH results in the suppression of testosterone production and spermatogenesis [12,14]. While this technique has been widely adopted in pigs, there is currently no licenced product in sheep [13,14,15].

Castration has been recorded as early as the 3rd millennium BC, while tail docking is less frequently described historically but thought to have become more widely used with the selection of sheep for longer and finer fleeces, which were more prone to accumulation of faecal matter and urine staining. Archaeological evidence of docking exists from the 13th century, and the procedure is thought to have become routine by the 18th and early 19th centuries; the agricultural revolution and popularity of the Merino breed played a major role in the widespread use of the technique [16].

The reasons for performing these procedures are historically much the same as they are today; castration is typically performed to prevent unwanted breeding, reduce aggression, improve stock person safety, and improve meat quality [12,17]. Castration techniques that remove the scrotum can also reduce the risk of flystrike and carcass contamination due to faecal matter building up on the scrotum [18]. Tail docking and mulesing are primarily performed to reduce dag formation (accumulation of faecal matter around the tail and hindquarter or breech) and urine staining in an effort to reduce the risk of breech flystrike [19]. Cutaneous myiasis, commonly known as flystrike, is the infestation of a wound by maggots and flies; it is a very painful condition and can result in death or significant morbidity [19]. These procedures can cause significant pain and distress associated with physical tissue injury, handling stress, and temporary separation from the dam (mother of the lamb) [20,21]. The age at which these procedures occur, the technique, and the analgesic or anaesthetic regimen vary between production systems and farms both within and between countries. All of these factors will influence the duration and severity of pain experienced by the animal. Our current understanding of the degree of influence these factors have on pain experience is limited. There is growing evidence from the human and rodent literature [22] suggesting longer-lasting effects that we have not appreciated as yet and have not been considered in legislative decisions. In the following sections, we will discuss the scientific evidence on the impact of different methods of performing common husbandry procedures, the ages at which they are performed, and associated pain mitigation strategies.

## 3. Animal Welfare Legislative Framework in Australia

Australia is a federation of six states and two territories (The *Commonwealth of Australia Constitution Act*
*1900* (*The Constitution*)), with laws at federal, state, and local government levels. The *Australian Constitution* (s 51) is silent on animal welfare, and thus, it is considered a residual power for which the eight Australian state and territory governments are responsible. The only exception to this is when animal welfare may be considered as part of an aspect of trade or biosecurity under the Federal government’s exclusive powers around trade and commerce and quarantine (s 51 (i), (ix))—a key example being regulation of the Australian live export trade.

As a result of these constitutional limitations, the Australian animal welfare legal framework consists of primary state and territory acts and delegated legislation. The former are overarching and general and provide the key offences. The main offences are a prohibition on being cruel to animals and the creation of a duty of care for owners of animals to provide for their welfare [23]. Subordinate legislation in the form of regulations, codes of practice, and standards are then used to provide greater technical detail on a species, type of production practice, or controversial issues [23]. Provisions written into regulations are usefully directly enforceable, with offences being directly associated with the provisions. These documents, therefore, have a greater legal weight than the so-called “soft” law or quasi-delegated legislation represented by Codes of Practice or Standards. Codes of Practice have a lower legal weight and ability to enforce, and their legal status varies considerably across jurisdictions. Their legal enforceability is also dependent on whether they are a compulsory or voluntary code of practice. For example, in South Australia (SA), a breach of a prescribed Code of practice provision is directly enforceable and subject to a penalty via Reg 5 (*Animal Welfare Regulations 2012*). Alternately, in some states, e.g., Victoria, compliance with a POCTA code merely provides a defence to a prosecution for cruelty under the enabling act. Voluntary codes in all states work similarly by assisting in the defence or prosecution of a cruelty charge in court. Codes may be incorporated into law by a variety of means. The most common method is either via direct referral in the regulations or by being listed as a prescribed code via a schedule (usually to the regulations, see, e.g., SA). A less common way of making them the law is through administrative means by referral in licence conditions around certain businesses. This method is commonly used in the regulation of animal slaughter [24]. From this brief background, the reader might already get the sense of how the different legal weight placed on these Codes can create a disparity between the jurisdictions with a scenario potentially existing where states are using the same document, but its enforceability varies due to its method of incorporation into the legislative framework.

### 3.1. History of Delegated Legislation around the Livestock Industries

During the 1800s, the states and territories introduced laws on animal welfare and animal cruelty offences based on equivalent regulations in Britain. At Federation in 1901, the states retained responsibility for those functions by virtue of the signing of the Constitution [25]. The 1960s saw the rise of animal rights advocacy, and the public’s attention was drawn to the conditions of animals used in intensive farming systems. The practice of mulesing lambs, debeaking chicks, and tail-docking piglets without pain relief was widely publicised [26]. The Australian export wool trade grew during the 1980s but saw a backlash from wool garment manufacturers and consumers against sheep that were mulesed. Australia’s sheep regulatory agencies aimed to develop consistent standards to reflect changing attitudes towards farm animal welfare [20,27].

Model Codes of Practice for the Welfare of Animals (MCOPs) were developed in the early 1980s with a focus on livestock. Their development was driven by the desire to provide consistent husbandry guidelines for all farm animals so that both domestic and international markets were assured of the welfare considerations made during the production of animal-sourced commodities [23]. However, in spite of good intentions to harmonise, each jurisdiction’s approach to the use of the Codes differed; some adopted them in their entirety, others modified them, whilst some chose not to adopt them at all [26].

These Codes were updated during the 2010s into the Australian Animal Welfare Standards and Guidelines (for all livestock species), overseen by the Primary Industries Ministerial Committee (PIMC) and in conjunction with each State’s department responsible for the Animal Welfare Act. Whilst the standards and guidelines are usually presented as one document, there is an important distinction between a standard and a guideline in these documents. The standards are the basis for developing consistent legislation across Australia and use the word “must”; hence, provided adopted by the states, they represent the legal requirements. Guidelines are recommended practices to achieve good welfare, and non-compliance will not constitute an offence under the law. Rather than eight different animal welfare regimes, the aim of these documents was to have [27]:


*“national standards of livestock welfare that are consistently mandated and enforced in all states and territories.”*


This national approach was to provide quality assurance and cost benefits for the primary industries and reflect modern animal welfare expectations from consumers [27].

In 2018, the Australian Productivity Commission (PC) issued a report on the regulation of Australian agriculture, including the management and welfare of farm animals [28]. It proposed the establishment of a federal Australian Animal Welfare Agency (AWAC) to oversee a nationally consistent approach to farm animal welfare and noted the importance of looking to scientific evidence and ethical values in setting industry standards. The Productivity Commission recommended the formal adoption of the Animal Welfare Standards and Guidelines endorsed by the Primary Industries Management Committee within each state and territory through the incorporation of these standards into their respective animal welfare laws [26]. However, this recommendation was not adopted federally, with the government’s response being to reiterate that the responsibility for animal welfare regulation, compliance, and enforcement fell to the state and territory regulators [28]. As a result, state Codes continue to vary in their recommended practices and currency.

### 3.2. Animal Welfare Laws and Codes of Practice for Sheep

Each state and territory has its own animal welfare or prevention of cruelty to animal acts [29], which enable either general or specific regulations and any compulsory Codes of Practice.

Subordinate legislation in the form of regulations is enabled under each act and is updated by the relevant responsible government department as necessary. As described earlier, these documents get their legal force via different mechanisms: direct referral or attached to schedules. Some states have elected to put Code or Standard provisions into their regulations directly to increase their legal weight. This is the scenario in SA where the sheep Standards and Guidelines have been incorporated into the regulations. New South Wales (NSW) and the Australian Capital Territory (ACT) have adopted the Standards and used the actual document, i.e., not amended it. Whilst Queensland (Qld) bases its own Code on the Standards. The Northern Territory (NT) does not have a code of practice for sheep. It is also worth noting the legal status of these documents. In the ACT, Victoria (Vic) and Tasmania (Tas), these are voluntary or advisory documents that are not directly enforceable but may be used for evidentiary purposes in court (Table 1).

## 4. Tail Docking and Castration—Science and Policy

Tail docking and castration are common husbandry practices in Australia [18]. The utility of tail docking is largely environment- and breed-dependent, and the need to carry out docking should be assessed based on unique climate and management conditions [36]. Similarly, the need for castration is based on management conditions and may not be required when lambs are marketed for slaughter prior to puberty, which typically occurs at 3–6 months of age [18,37].

There is conflicting evidence linking undocked sheep with an increased risk of breech flystrike [36,38,39]. Scobie et al. [40] found that dag formation was dependent on seasonal and management factors, and the length of the tail did not alter the risk of flystrike. Watts and Marchant [41] compared lambs docked either at the third palpable tail joint or as short as possible, finding that flystrike was far more common in short-docked sheep. Flystrike is more common in warm and wet weather, particularly in breeds with wrinkled skin and wool- or hair-covered breech [19,42]. Other factors, such as parasite burden and nutritional imbalance leading to diarrhoea, can increase the risk of flystrike [38].

Management of these factors plays an important role in the prevention of flystrike. In regions where there is a higher risk of flystrike, tail docking at the third palpable joint has been associated with the least amount of dags and urine staining in ewes [40,43]. Tail docking at the third palpable joint, which is equivalent to the length of the vulva, is also widely recommended as the optimal length to reduce the risk of vulval cancer, bacterial arthritis in lambs [44], and rectal prolapses and to maintain rectococcygeal muscle integrity [43,45].

Tail docking and castration are commonly studied in conjunction and are discussed together. Surgical castration and tail docking are rarely used in Australia. In 2016 only 3% and 6% of surveyed Australian sheep producers still used a sharp knife for castration and tail docking, respectively [46]. Surgical methods are associated with a higher risk of complications, such as haemorrhage, compared to bloodless methods, such as the use of rings. There is also evidence of a greater physiological stress response with surgical castration and tail docking, with these procedures causing a greater and more prolonged increase in cortisol compared to rubber ring castration and tail docking [47,48,49]. Mellor et al. [50] conducted a review of castration and tail docking techniques, using cortisol response to rank the severity of commonly used techniques, including surgery, rubber ring, and hot iron tail docking with and without analgesia. They recommended that surgical methods be phased out in preference of ring methods, ideally with local anaesthetic instilled prior to ring application [50]. In contrast to this, a number of studies assessing behavioural signs of acute pain following husbandry procedures indicate that castration and tail docking with rubber rings causes a greater and more prolonged negative welfare impact compared to surgical methods [21,51,52,53,54,55]. Lomax et al. [54] used nociceptive threshold testing to compare wound sensitivity in 6–12-week-old Merino lambs that had been surgically castrated with or without topical anaesthesia, revealing significant primary and secondary hyperalgesia for at least 4 h after the procedure in the castrated lambs that had not received analgesia. Allodynia around the castration site was not identified in any of the lambs. In this study, lambs that were tail docked with a sharp knife and received no analgesia developed allodynia at the tail wound site 4 h after the procedure, and all surgically docked lambs, regardless of analgesia, developed primary hyperalgesia at the wound site. Lamb’s tail docked with a hot iron showed no evidence of primary hyperalgesia or allodynia up to 4 h after docking. The application of a local anaesthetic to the hot knife wound reduced tail wound sensitivity from baseline levels [54]. Analysis of acute pain-related behaviours between groups of lambs that were either surgically castrated and tail docked, surgically castrated and hot knife docked, or rubber ring castrated and tail docked demonstrated a marked increase in behaviours in the ring group that dominated their experience to the point where nociceptive threshold testing could not be performed. There was no significant difference in pain-related behaviours between handled control lambs and lambs that were surgically docked or docked with a hot iron and had local anaesthetic applied at the time of the procedure [54].

Other bloodless methods of castration include use of various clamping instruments to crush the spermatic cords and testicular blood supply, inducing ischaemic necrosis. One of the commonly studied castrators is the Burdizzo. The use of this device in combination with the rubber ring reduced the length and duration of behavioural signs of pain and cortisol response in lambs compared to ring and surgical castration [50,56,57]. Despite strong evidence that the combination of Burdizzo castrators applied proximally to rubber rings reduces pain following castration, they have not been adopted widely because they are technically difficult to use, there is a higher risk of procedural failure, and they increase the time taken for marking [58].

There is strong scientific evidence that local anaesthetics injected subcutaneously or applied directly to the wound reduce behavioural and physiological sings of acute pain following tail docking and castration in lambs from 2 days up to 12 weeks of age [59,60,61,62]. Non-steroidal anti-inflammatories (NSAID) have also been shown to reduce some pain behaviours and physiological signs of pain following castration and tail docking [63,64]. Small et al. [65] found significantly lower lamb mortality from marking to weaning in ring-docked and castrated lambs treated with the NSAID meloxicam compared to those that received no form of analgesia. This study did not have a handled control group, and the causes of lamb mortality were not recorded, which limits the conclusions that can be drawn. However, lamb losses are a significant welfare issue and economic burden in the sheep industry, and further investigation of this finding is warranted.

Both local anaesthetics and NSAIDs reduce acute pain but do not completely ameliorate it, and they do not address chronic pain associated with tail docking and castration [66,67]. Hyperalgesia at the tail docking site following hot knife docking can last for at least 3 months [67]. Currently, there are no pain mitigation options addressing the chronic component of pain associated with these tail docking and castration.

A 2016 survey of Australian sheep farmers found that 97% of producers use rubber rings for castration. The same survey found that the selection of tail docking method changed depending on the production system, with most wool producers electing to use a gas knife (78%), whereas meat producers tended to prefer rubber rings (65%). However, it is likely that there are also some state-by-state differences, with gas knives being more commonly used in WA and SA (74% and 75%, respectively) compared to other Vic, Tas, Qld, and NSW (45%, 59%, 33%, and 49%, respectively). Qld farmers reported the highest proportion of farmers using a sharp knife (28%) [46].

### 4.1. Tail Docking Policy

In considering policy around tail docking, there is inconsistency across the jurisdictions around the need for a certain length of tail. This may reflect the difference in minimum standards and guidelines advised by the AHA Animal Welfare Standards and Guidelines [30]. The Qld and SA codes state the required minimum standard, which is to leave a tail stump of at least one palpable joint, whereas the Tas, WA, Vic, and NSW codes require tail length to be long enough to cover the vulva in ewes and be a similar length in males, which is equivalent to three palpable joints. This fits with the scientific evidence around tail docking length and is in the guidelines advised by AHA [30]. There is also inconsistency in the recommended age for procedure performance, with some states not providing any guidance on this other than the age at which anaesthesia must be used. The guidelines recommended by AHA advise tail docking to be performed as early as possible and before 12 weeks of age. Of particular interest in relation to this is that SA and Qld, states that have adopted the Standards and Guidelines, appear to have made a deliberate omission of this recommendation in their laws. The recommended age for tail docking across most of the states is between two and twelve weeks and not until at least 24 h old to allow for parental bonding. It is also worth noting that the newer standards and guidelines endorse the performance of this procedure at an earlier age (from 24 h) in comparison with the older state codes from Vic and WA. Pain relief and/or anaesthesia are only *required* for lambs over six months of age (Table 2). The guidelines recommend that suitable pain relief is used when practical and economically feasible despite strong scientific evidence that tail docking is painful, even in very young lambs [51], and that analgesia mitigates acute pain following tail docking [52,54,60,63]. Other than the requirement of anaesthesia in sheep over 6 months of age, there is no recommendation to use analgesia for younger lambs in any of the state codes (Table 2).

### 4.2. Castration Policy

Most codes recommend castration take place as early as possible, generally between 24 h and 12 weeks. Analgesia or anaesthesia is required only if the ram is over six months. Recommended methods are either by rubber rings or cutting, although the Standards and Guidelines suggest an “appropriate tool that causes the least pain” (Table 3). This flexibility in the choice of the method provided by policy likely reflects the controversy around the relative welfare impact of the methods, with no method conclusively being shown to have less animal impact. South Australia’s Regulations do not specify a recommended age or method, again an interesting observation since this State has adopted the Standards and Guidelines which are not silent on these matters.

## 5. Mulesing-Science and Policy

Mulesing was developed in Australia in the late 1920s by a grazier called John Mules. The procedure involves surgically removing the wool-bearing, wrinkled skin around the perineal region to enlarge the bare area of the breech and prevent the build-up faeces and urine in the wrinkles, thus reducing the risk of breech flystrike [20,68]. At the time of development, the Australian Merino sheep industry was struggling with a significant increase in flystrike-associated morbidity and mortality due to the breed’s wrinkled breech, favourable Australian weather conditions for flies, and the introduction of the fly *Lucilla cuprina* [20], which accounts for at least 90% of all strikes [69]. Mulesing was a cheap, fast, and effective method of reducing the risk of breech strikes, and the popularity of the procedure gradually grew, with 70% of Australian Merino producers mulesing their ewe lambs in 2017 [70]. Flystrike remains a significant issue for the Australian sheep industry, costing just over $323 million AUD in prevention, treatment, and production losses annually [69]. It is worth noting that this is a particular issue with the prevalent Merino breed used in Australia for wool production due to the amount of wrinkling. If other breeds were used, the problem would no doubt be reduced. The procedure has received global scrutiny for its negative impacts on welfare and is banned in most countries, with our close neighbours New Zealand banning the procedure in 2018 [71]. Phasing out mulesing in New Zealand was largely industry-led and took roughly 5 years [72]. Differences in the Australian climate, wool industry, predominance of the Merino, and larger enterprises compared to the New Zealand wool industry have substantially delayed the phasing out of mulesing in Australia [72]. Nevertheless, Australian livestock industries are working towards phasing out mulesing through research into alternative ways of preventing flystrike, including breeding programmes to reduce wrinkle scores [21,68] and developing extension strategies to educate and support producers transitioning to non-mulesing operations [72]. Flystrike remains a major concern, and the risk of flystrike is expected to increase with the emergence of chemical resistance, limiting the efficacy of chemicals used for prevention and treatment [68,69]. It has also been suggested that the distribution and abundance of the fly population may increase with climate change [68], thus increasing the risk of flystrike. This highlights the importance of continued support for research investigating flystrike prevention and treatment to foster a sustainable sheep and wool industry that is able to meet consumer demands and maintain the social license to operate [72].

Producers commonly use mulesing shears to remove the skin around the breech and on either side of the tail, leaving an open wound that heals by secondary intention leaving a wool- and wrinkle-free area [20]. This procedure is widely known to cause considerable pain that can persist for days to weeks [73,74,75,76]. Surgical mulesing elicits marked changes in physiological and behavioural markers. Behavioural indicators of pain after mulesing include statue standing, hunched posture, reduced lying behaviour, longer time to mother up and feed, reduced grazing behaviours, and an aversion to the handler [55,74,75,77,78,79,80]. Significant elevations in cortisol have been demonstrated in lambs aged from 5–40 weeks of age [73,74,75,76,81]. Other physiological markers of inflammation and pain, including haptoglobin, neutrophil/lymphocyte, and beta-endorphins increase dramatically [73,74,75]. Likewise, numerous studies have demonstrated a reduction in average daily weight gain for the week following mulesing [75,76,77]. While the use of analgesics reduces the physiological effects and behavioural aberrations associated with surgical mulesing, they do not abolish them and do not address any chronic effects [82]. A combination of a non-steroidal anti-inflammatory drug (NSAID) and topical anaesthetic and antiseptic has been shown to provide the most effective pain relief [83]. Registered products available for mulesing in Australia are Metacam^®â^, an injectable form of the NSAID meloxicam, Buccalgesic^®^, an oral formulation of meloxicam, and Tri-Solfen, which is a gel-based spray containing lignocaine, bupivacaine, adrenaline, and cetrimide. Administration of Tri-Solfen^®^ and either form of meloxicam at the time of mulesing significantly reduced pain-related behaviours over the first 24 h post-procedure [80,83]. A multimodal approach to analgesia has been shown to be superior to the use of sole agents [81,83,84].

Various non-surgical alternatives to mulesing have been trialed with little or no improvement in welfare or flystrike prevention over the surgical mulesing technique. These include clips to induce ischaemic necrosis around the breech and later sloughing of the tissue, injection of chemical agents (sodium lauryl sulphate, cetrimide) to induce scar formation around the breech or application of liquid nitrogen to the breech resulting in necrosis of the skin [75,76,77,78,85]. Other strategies of flystrike control, such as breeding for reduced breech wrinkling, preventing scouring, management practices including regular crutching and appropriate use of insecticides, and managing the fly population are more viable long-term options that will maintain the social licence and the marketability of Australian wool and meat products globally [19].

Most states require lambs to be over 24 h old before mulesing and recommend between two and twelve weeks of age (Table 4). In spite of the common recommendation for the procedure to be performed between 2 and 12 weeks, it is noteworthy that all states essentially allow the procedure in animals up to the age of 12 months, with WA having no upper cap on age. This large window for performance may reflect the practicalities of performing this procedure, for example, in acquiring the services of accredited mulesing contractors. In some states, there is guidance around operator competency with detailing around knowledge/experience required or supervision needed. Only Vic requires that operators have received some form of accredited training in order to perform the procedure. Victorian guidelines also require analgesia for all sheep, plus anaesthetic if over 6 months, although it is important to remember that this is a voluntary code, so this may not actually be the routine practice in this state (Table 4).

## 6. Veterinary Legislative Framework

Whilst the veterinary legislative framework is not focussed on welfare, it may provide useful guidance around the perceived severity of these procedures. These procedures are listed within the veterinary legislative framework in all states (with the exception of Vic) in the context of defining an act of veterinary medicine or surgery (Table 5). In all states, the performance of an act of veterinary medicine without being a registered veterinary professional constitutes an offence under the Act. Across the states, these two frameworks are generally consistent in terms of legality, e.g., if a person complies with one framework, they are unlikely to fall foul of the other, perhaps with the exception of SA (see table for detail). In general, the veterinary frameworks are less conservative with respect to age limits on procedures; there is a broader range of early ages considered before they become acts of veterinary surgery. This is not unsurprising as these documents are not aimed at farmers and do not dictate routine husbandry practices. However, they show broad consistency across the states with the proposition that castration and tailing of sheep over 6 months and mulesing over 12 months represent a procedure that should be performed by a veterinarian. This points to an appreciation that by this age, a more advanced level of surgical skill may be required, and of course, access to analgesic or anaesthetic drugs.

## 7. Discussion

Regularly reviewing the need for these procedures and available alternatives will benefit not only animal welfare but also the economics and efficiency of the relevant farming enterprises. This is particularly important in the current climate, where the welfare impacts of these procedures are being challenged by consumers and the wider public. The long history and culture surrounding these procedures result in skills and techniques being passed down across generations, sometimes with little change or consideration of newer techniques or improvements. A survey of Australian sheep farmers found that 20% of farmers stated that their father had played a substantial role in their attitude towards animal welfare [6]. This transfer of skills is highly valued within farming communities and should be respected and considered as recommendations and standards are updated. It is important to also consider a science-based approach to guideline formation, and indeed transparency of this approach, i.e., it is clear to reviewers what the scientific basis of the recommendations is. This is a key component of clinical practice guideline formation in evidence-based medicine, and it is surprising that clear referencing of the science has not been more extensively adopted into the often-controversial animal welfare policy space. A brief discussion based on our observations on the linkage between science and law in this policy area follows.

### 7.1. Age at Marking

Recommendations on the age at which husbandry procedures should be performed in lambs vary moderately across Australian state and territory legislation. However, recommended age limits globally are considerably different [94]. This is likely to lead to some confusion among producers and the public, particularly in relation to international trade. There are three ages that are commonly cited in legislation internationally as limits for performing painful husbandry procedures: less than 7 days old, less than 12 weeks old, and less than 6 months old. For example, rubber ring tail docking is only permitted up to 7 days of age in England [95] and Wales [96], whereas in Canada, it is recommended to be performed before 7 days of age and prohibited over the age of 6 weeks [97]. In Scotland, tail docking can be performed up to 3 months before veterinary oversight is required [98]. New Zealand recommends tail docking be performed before 6 weeks of age [71]. This variation appears to reflect the prominent production system in the relevant regions but fails to reflect the scientific reasoning behind such choices. Consideration of the production system is, of course, vital to creating recommendations of practice, as often these systems have developed over the course of centuries and techniques have been learnt and passed down through generations in response to the local environmental and cultural factors. A 2016 survey of Australian sheep farmers found that most lambs were castrated and tail docked at an average of 6.5–6.7 weeks. No farmers surveyed reported docking over 6 months and only 5% of farmers reported docking lambs over 3 months [46]. This reflects the common Australian practice of mustering all ewes and their lambs together for marking at one timepoint after the end of lambing when lambs are all at least 2 weeks of age, thus reducing the stress of repeat handling. Here, we discuss the research exploring the influence of age on response to husbandry procedures and how this may inform the legislation listed in the previous section.

Research comparing tail docking and castration in lambs of different ages has repeatedly demonstrated a significant increase in pain behaviours and physiological measures of pain across all studied age ranges [47,50,51,66,99]. Kent et al. [47] and Molony et al. [51] measured cortisol and behavioural changes acutely after rubber ring, surgical, or rubber ring and burdizzo castration in 5-, 21-, and 42-day-old lambs. In all ages, all methods caused a significant increase in cortisol from baseline, with the peak occurring earlier in surgical and ring with burdizzo methods and roughly 10 min later in rubber ring castration and tail docking. The change in cortisol was significantly greater in the 5-day-old lambs after rubber ring castration compared to the older lambs. Kent et al. [66] compared active pain behaviours, and scrotal lesion width and degree of swelling following castration with a rubber ring in 2-day and 28-day-old Suffolk or Dorset cross lambs and 42-day old Scottish Blackface lambs to assess chronic inflammatory responses and long-term pain of this procedure. In the younger lambs, the scrotal lesions were smaller and healed more rapidly than in the 42-day-old lambs, and they were less likely to become septic. There was a significant relationship between the increase in active pain behaviours and the change in lesion score and size only in the 42-day-old lambs and not the 2-day and 28-day-old lambs [66], suggesting that the older lambs tended to suffer from larger and more painful lesions than the younger lambs. Electroencephalography has provided another method of assessing pain in lambs following husbandry procedures. Using electroencephalography while lambs were under halothane anaesthetic, Johnson et al. [100] and Johnson et al. [101] found that older lambs had a more pronounced cerebro-cortical response to castration with a rubber ring than younger lambs. In the first 10 days of life, the magnitude of the cortical response to noxious stimuli (castration with a rubber ring) increases rapidly [101], suggesting that the perception of noxious stimuli in younger lambs is different and likely less pronounced compared to older lambs. However, these studies only assessed male lambs and pain perception, and the consequences of early life pain may differ between sexes [102,103]. Whilst it generally appears that procedure performance at an earlier age is beneficial, there is a growing body of research demonstrating longer-term effects of painful procedures during early development in a range of species, including lambs [104,105,106], humans [107], and rodents [103,108]. These findings point towards the need to consider pain relief in animals of all ages to avoid later negative consequences.

Our understanding of the perception of pain in infancy in humans has changed significantly over time. Historically, there was a general acceptance that infants felt pain, and efforts were made to alleviate that pain by ancient physicians and philosophers; this social dogma changed to a denial of the clinical significance of infant pain which lasted throughout most of the 20th century [109]. It was thought that babies and young animals did not perceive or suffer from pain in the same way adults do due to an underdeveloped nervous system. Additionally, the risk of anaesthetics and analgesics in these patients was seen to outweigh the seemingly limited benefits. Consequently, numerous painful surgical procedures were performed without any form of anaesthetic or analgesic [110]. Further research has revealed poor pain management during early life can have a range of negative repercussions on pain sensitivity, cognition, social interaction, mood, and stress resilience in later life [94,109]. There is also evidence of intergenerational effects of early-life pain in rodents [111] and sheep [106]. Considering these negative consequences of early life pain identified predominantly in rodents and humans, it is worth investigating further in other species. Farmed species, such as sheep, are exposed to numerous painful procedures very early in life and are often at a relatively high risk of infection and inflammation due to their outdoor or intensively housed environment. Theoretically, the negative consequences of early-life pain and inflammation would be expected to be present in these populations. There are some studies investigating the influence of painful husbandry procedures in sheep and cattle on later-life pain and productivity. Clark et al. [106] demonstrated increased pain behaviours during parturition in 2-year-old ewes that had been exposed to lipopolysaccharide (LPS) at 48–72 h old to simulate a mild infection and ewes that had been tail docked at 72–96 h old compared to controls. Those ewes exposed to LPS also had a significantly longer inter-birth interval than control ewes, suggesting some influence on birth ease through currently unknown mechanisms. This study went further to measure nociceptive thresholds of lambs from these ewes during tail docking at 3 days of age, finding lambs from the LPS treated group had significantly higher mechanical nociceptive thresholds across 2 days compared to lambs from tail docked ewes and controls. Altered pain perception later in life following injury or illness at an early age was also demonstrated by McCracken et al. [104], who found that male lambs castrated with a rubber ring at 1 day old displayed significantly greater pain behaviours at tail docking with rubber ring at 26–34 days old, compared to lambs that had been castrated at 10 days of age. These studies in sheep are consistent with findings in the human and rodent literature, suggesting that painful procedures early in life negatively affect pain perception and may make affected individuals less resilient to later life stressors. This leads us to question the relevance of the legislation implying that if painful husbandry procedures are performed at an early age (which varies depending on location), pain mitigation strategies are deemed unnecessary. Granted, the methods used to perform tail docking, castration, and mulesing are best performed at a younger age to reduce the size and developmental complexity of tissue affected [66], but this does not override the need for appropriate analgesia to be provided in all cases.

From a practical approach, the current analgesic options available to farmers are somewhat limited as they have a relatively short duration of action (30 min to 72 h [82,112,113]), peri-operative analgesia or anaesthesia requires prior administration and double-handling, they can also be impractical or difficult to administer during marking, and they can be cost prohibitive. There is also evidence that education on the use of pain relief is lacking, as a number of surveys show that producers are not always using available analgesics appropriately [112,114]. For example, a survey of Australian sheep producers found that of the 30% of producers using pain relief for rubber ring castration, over half (58%) reported using Tri-Solfen^®^, which is an unsuitable analgesic for this method and indicates a misunderstanding of the mode of action of this product [112]. Additionally, the use of suitable multimodal analgesia (NSAID and appropriate local anaesthetic), which is the current best practice [82,83], was used by less than 10% of producers for castration and tail docking (1% and 7.7%, respectively) [112]. Education programmes covering the recognition of pain in animals, the detrimental effects of pain, and the use of pain relief are clearly an important part of promoting the appropriate use of pain relief for husbandry procedures [8,45,112]. Effective dissemination of new scientific findings and legislative changes or recommendations is crucial for constructive development within the farming sector. Government departments and agencies are not always seen as trusted sources of information [45,115]. Whereas experienced farmers are seen as trusted advisors within farming communities [45], and knowledge and experience are often passed down between farmers and families through informal training [45,46]. These factors should be taken into consideration when developing intervention and education strategies to effect real change.

In spite of this discussion, it is, however, heartening to see that whilst only 8.4% of Australian wool producers who routinely mules their lambs and provide pain relief reported using optimum combination analgesia, 92% of producers who mules do use some form of pain relief [112]. This is despite policy only requiring it over 6 months. This finding serves to remind us that the law is merely there to set a minimum standard, a level playing field, as it were. Farmers can, and clearly do, practice at a higher standard spurred on by industry guidance or incentivisation via assurance schemes.

### 7.2. Farmer Perception

Farmer perception of routine husbandry procedures often differs significantly from the public’s [9,116]. Soriano et al. [9] surveyed Brazilian sheep producers and the public about their impression of animal welfare issues in sheep farming and their knowledge of animal protection laws. Only 3.7% of the farmers that performed tail docking used an anaesthetic for the procedure, but 45.7% of the surveyed farmers stated that this was a form of animal maltreatment. This contrasts with 88.9% of citizens who thought this was maltreatment. Another interesting finding from this survey was the limited awareness of animal protection law; only 5.9% of farmers knew of the laws, but they could not cite any, whereas significantly more citizens (17%) knew of the animal protection laws. Similarly, Woodruff et al. [45] found that a lack of awareness of the recommended tail docking length was a major factor driving docking practices in 57% of surveyed Victorian sheep farmers that docked tails shorter than three palpable joints. Knowledge and implementation of current legislation and guidance on farming practices among the farming community appear to be a major barrier globally [7,9,45,46]. This leads us to query the efficacy of enforced legislative changes over other methods of knowledge dissemination and changing practice, such as education programmes for farmers and other stakeholders. Ultimately, a combination of legislative change and stakeholder-led education programmes is more likely to create sustained and widespread improvements in animal husbandry. The disconnect between industry and citizen viewpoints also poses a key challenge for legislators who must balance multiple viewpoints and priorities when making public interest laws (such as animal protection laws).

### 7.3. The Science-Policy Interface in Animal Law

It is generally considered that the drafting of law takes into consideration the prevalent scientific evidence. Law reform bodies also commonly commit to driving evidence-based law reform processes as well as enhancing the decision-making around policy inclusions considering both transparency and democratisation of the processes [117]. This inclusion of science into law has clear potential benefits for all major stakeholders: for the animals, it is hoped that the use of welfare science will ensure policy that at least safeguards their welfare, if not improves it; for citizens and consumers, it should serve to reassure them of this welfare protection, and for the industry, it should provide uniformity of practice across jurisdictions creating a level playing field which is particularly relevant in matters of trade. However, it is naïve to think that science will completely inform the content of written law; instead, the law likely reflects a delicate balance between competing interests, viewpoints, and topical societal opinions. In the area of food law, three approaches have been used to describe regulation in this area. These approaches likely hold similarly for animal welfare law. These have been labelled “political—democratic”, “economic”, and “scientific” [118]. In a political–democratic approach legal content is determined by the support that the majority will lend to a certain opinion, i.e., the public determines the law. As the name suggests, in the economic approach, the law is driven by economic forces, whereas the scientific approach leaves the experts (scientists) to decide the legal provisions. In reality, these approaches likely overlap in the law drafting and consultation phases. However, at their intersection, there is often an inherent tension between scientists who tend to pursue objectivity and the elimination of bias and the process of policymaking. The latter requires consideration of objective information (guided by science) and subjective value judgements (such as the nature of the desirable outcome or the balance of competing interests). This often lends to the scenario when people may agree on a common set of facts but disagree on the appropriate policy response [118]. It is also worth noting that, at least in respect of animal welfare law, science may play a greater or lesser role depending on the nature of the subject matter and the positioning in the regulatory framework. As an example, there is probably a greater opportunity for incorporation of science into technical material on animal husbandry and use in delegated legislation, for example, around tail docking. However, prevailing societal viewpoints may play a greater role when considering broad provisions around animals in general, for example, around the inclusion of sentience or whether certain practices are justified. In consideration of methods of driving law reform, it is also important to consider that law reform does not occur in a vacuum; laws generally follow community attitudes rather than shaping them, and as a result, legal reform needs support from a broad community base [119,120].

Notwithstanding the need to balance science against other societal and economic considerations in legal drafting, a further challenge for policymakers is how to source and evaluate the welfare science available and its value for inclusion. Jurisdictions typically approach this using different methods, which may be influenced, at least in part, by resources available. It is possible that it is the reliance on different methods of assessing and critiquing the relevant science that contributes to the diversity in legislation in this area.

The European Union utilises a multi-step approach to the incorporation of science into law. The EFSA (European Food Safety Authority) is an agency of the EU with its core activity being to collect, appraise and integrate scientific evidence to address risks [121]. One of EFSA’s panels is dedicated to Animal Health and Welfare. This committee is made up of European scientists with expertise relevant to animal welfare and health. A key feature of the EFSA groups is their commitment to independence with strict working practices to reduce conflict of interest as well as members being vetted for any conflicts of interest, which may include the provision of advice or services to any industry covered by EFSA’s work. Typically, this group produces substantial reviews of the scientific literature on the topic of interest, incorporating a risk of bias assessment and assessment of certainty in the evidence. The culmination of this work can then be used by policymakers at both the EU and state level to feed into law reform [122] via a standard democratic process involving consideration of stakeholders generally achieved via representation of the member states in the EU Parliament [123]. Additionally, as is common with the law-making process, if any policy is expected to have a considerable impact economically, socially, or environmentally, an impact assessment will need to be prepared to gauge the impact on stakeholders [123]. The EU system is arguably unique in providing considerable resourcing of scientific expertise to contribute to the law-making and implementation process. The focus on the provision of independent and non-biased advice, along with the assessment of certainty in the scientific evidence based on established principles of evidence-based practice, is also laudable. Other jurisdictions adopt facets of this model. For example, when the new Standards and Guidelines were generated in Australia, a review of the literature was used in development of the Australian Animal Welfare Standards and Guidelines for Pigs. Whilst this review was funded by the Australian Pork Industry, it was subject to independent peer review [27]. A comprehensive independent review of Cephalopod Molluscs and Decapod Crustaceans was also recently commissioned by the UK government prior to recommending the inclusion of these species as “animals” in animal welfare law [124]. However, it appears that in most jurisdictions, the approach to performing reviews of the literature is somewhat ad hoc and perhaps based on perceptions of risk due to public interest in the area. There is also variability in the types of reviews performed with varying use of systematic methods to incorporate an evaluation of certainty in the evidence to guide policy-makers.

Whilst outside the scope of this article, it is also worth mentioning that law-making in this area may be subject to regulatory capture [125]. This is defined when a regulatory agency is acting in the interests of the industry it is regulating and, in doing so, is creating an inconsistency with the public interest. This may particularly be a risk when there is an overrepresentation of industry in standards/code development and when there is industry control over the direction and reporting of welfare science conducted through channelling of funding [125]. The EU process of independent expert review of the science may go some way to avoid regulatory capture but is unlikely to have full effect; this likely requires considerable procedural change at both the law-making and enforcement levels.

## 8. Conclusions

There is broad consistency across the Australian jurisdictions in relation to specific provisions around procedures at lamb marking, for example, recommended age ranges at the performance of procedure and the use of anaesthesia in animals over 6 months old. Recommendations for the use of pain relief in animals less than 6 months are non-specific or absent in most states. The scientific evidence surrounding marking procedures indicates that castration, tail docking, and mulesing cause pain acutely and for at least two days post-procedure, regardless of age. This pain can be mitigated to some extent by NSAIDs and local anaesthetics that are licensed for use in sheep in Australia. There is a clear disconnect between the relevant legislation and scientific evidence. Legislation is lacking in traceability back to the science through no direct referencing. It also remains unclear from documents in the public domain what the scientific basis was and the process for assimilating the science of sheep husbandry during the creation of the new sheep standards and guidelines.

In spite of calls for national harmonisation of requirements around farm animal welfare, there is obvious dis-uniformity between the states and territories around provisions related to sheep. This has arisen through inconsistent incorporation of the new Standards and Guidelines into the state’s welfare frameworks, with some states either retaining their own document or modifying the published version. Moreover, the enforceability of the Code varies considerably across the states. Vic provides an excellent example of this with a Code that arguably is the most welfare friendly by requiring all mulesed animals to have pain relief and for operators to be formally trained in the procedure. However, this code is a voluntary code of practice in this state, and therefore, there are no direct ramifications for failing to adhere to these provisions. In comparison, NSW, having adopted the AHA Sheep Welfare Standards and Guidelines, requires that pain relief be used in sheep that are mulesed between 6 to 12 months of age. These standards are mandatory, and violation of them may be used as evidence of an offence in court [126].

Whilst resourcing is likely to be an issue, there is a need for Australia to address how well its animal welfare policy documents reflect current scientific evidence, as well as the transparency of this incorporation. A consideration of the processes and people involved in the making of delegated legislation in this area of public interest law is needed. This need is especially urgent given enhanced global trade networks, and current scrutiny of our practice brought about through recent free trade agreements.

## Figures and Tables

**Table 1 animals-13-01358-t001:** State Codes of Practice for sheep and year of publication.

Jurisdiction	Code of Practice	Voluntary or Compulsory	Year of Publication
ACT	Code of Practice for the Welfare of Animals- Sheep (uses the Australian Animal Welfare Standards and Guidelines—Sheep [30])	Voluntary	2016
NSW	Australian Animal Welfare Standards and Guidelines—Sheep [30]	Compulsory	2016
NT	None		N/A
QLD	The Code of Practice about Sheep [31] (based on the Australian Animal Welfare Standards and Guidelines for sheep)	Compulsory	2021
SA	No code of practice per se. See offences provisions in Part 9 of the Animal Welfare Regulations 2012 [32] (incorporated Standards)	NA	2017
TAS	Animal Welfare Guidelines—Sheep [33]	Voluntary	2008
VIC	Code of Accepted Farming Practice for the Welfare of Sheep (Victoria) [34]	Voluntary	Revision No. 3—no date
WA	Code of Practice for Sheep in Western Australia (2003) [35]	Compulsory	2003

**Table 2 animals-13-01358-t002:** Comparison of the recommended age, method, and pain relief for tail docking in sheep according to each jurisdiction’s code of practice for the welfare of sheep. * Noting that the detail is in the regulations in SA.

Jurisdiction	Recommended Age	Method of Tail Docking	Analgesia or Anaesthesia Requirement
ACT	Between 24 h and 12 weeks	Hot knife or rubber ring	If over 6 months, pain relief is required.
NSW	Between 24 h and 12 weeks	Hot knife or rubber ring	If over 6 months, pain relief is required
QLD	Not specified	If under 6 months, method must avoid unnecessary pain or suffering	If over 6 months, pain relief is required
SA *	Not specified	Not specified	If over 6 months, an anaesthetic or analgesia is required
TAS	As early as management practices allow	If without anaesthesia, a sharp knife, rubber ring, or searing iron	If without anaesthetic, as early as possible, preferably before 12 weeks, and not over 6 months
VIC	Between 2 and 12 weeks	If without anaesthesia, a sharp knife, rubber ring, or scarring iron	If over 6 months, an anaesthetic is required
WA	Between 2 and 12 weeks	If without anaesthesia, a sharp knife, rubber ring, or searing iron	If over 6 months, an anaesthetic is required

**Table 3 animals-13-01358-t003:** Comparison of the recommended age, method, and pain relief for castration in sheep according to each jurisdiction’s code of practice for the welfare of sheep.

Jurisdiction	Recommended Age	Recommended Method	Analgesia or Anaesthesia
ACT	Between 24 h and 12 weeks	Appropriate tools that cause the least pain	If over 6 months, pain relief is required
NSW	Between 24 h and 12 weeks	Appropriate tools that cause the least pain	If over 6 months, pain relief is required
QLD	Not specified	Method must avoid unnecessary pain or suffering. Ram must be over 6 months if the cryptorchid method is used	If over 6 months, pain relief is required
SA *	Not specified	Not specified	If over 6 months, an anaesthetic or analgesia is required
TAS	As early as management practices allow	If without anaesthesia, cutting, rubber rings, or emasculators/spermatic cord crushing instruments	If over 6 months, an anaesthetic is required
VIC	As early as management practices will allow, preferably before 12 weeks	If without anaesthesia, cutting, or rubber rings	Not specified
WA	As early as management practices will allow, preferably before 12 weeks	If without anaesthesia, cutting, or rubber rings	If over 6 months, an anaesthetic is required

* Noting that the detail is in the regulations in SA.

**Table 4 animals-13-01358-t004:** Recommended age and requirement for analgesia or anaesthetic during mulesing procedure for each state.

Jurisdiction	Recommended Age	Analgesia or Anaesthesia	Associated Conditions
ACT	Between 2 and 12 weeks (guidelines), but not less than 24 h or over 12 months.	If over 6 months, pain relief is required	Person mulesing must have the relevant knowledge, experience, and skills or be under the direct supervision of such a person Skin must not be removed unless it is wool-bearing skin
NSW	Between 2 and 12 weeks (guidelines), but not less than 24 h or over 12 months.	If over 6 months, pain relief is required Skin must not be removed unless it is wool-bearing skin	Person mulesing must have the relevant knowledge, experience, and skills or be under the direct supervision of such a person Skin must not be removed unless it is wool-bearing skin
QLD	Not less than 24 h nor over 12 months	Not specified	Skin must not be removed unless it is wool-bearing skin
SA	Not less than 24 h nor over 12 months	If over 6 months, an anaesthetic or analgesia is required	Skin must not be removed unless it is wool-bearing skin
TAS	Between 2 and 12 weeks, but not over 12 months.	If over 6 months, an anaesthetic is required	Mulesing should be done in conjunction with lamb marking to minimise stress and handling
VIC	Between 2 and 12 weeks. If over 12 weeks, only in exceptional circumstances.	Pain relief is required for all mulesed sheep. If over 6 months, an anaesthetic is required.	Legislation in States and Territories covering regulation of veterinary procedures and/or animal welfare must be complied with Persons carrying out the mulesing procedure must have appropriate competencies, demonstrated following a formal accreditation process or by other assessment by a Registered Training Organisation.
WA	As soon as possible after 2 weeks of age	Not specified	Where possible, perform in conjunction with other lamb marking operations.

**Table 5 animals-13-01358-t005:** Veterinary legislative framework across the states and territories with respect to marking procedures.

Jurisdiction	Legal Provision	Relevant Provisions	Level of Agreement with Animal Welfare Legislative Framework
ACT	*Veterinary Practice Act 2018* *Veterinary Practice Regulation 2018 [86]*	Castration or tailing of a sheep older than 6 months and mulesing over 12 months are acts of veterinary science (part 1.2 regulations)	Consistent with animal welfare framework recommending performance up to 12 weeks for docking and castration, with veterinary surgeons presumably performing at a later age since s 10 Act makes an offence to carry out a restricted act of Veterinary science.
NSW	*Veterinary Practice Act 2003* *Veterinary Practice Regulation 2013 [87]*	Castration and tailing of sheep older than 6 months and mulesing over 12 months of age are restricted acts of veterinary science	There is consistency between the legislative frameworks, with the Standards and guidelines being more conservative in relation to age, i.e., recommending performed at a younger age. Act s 9 makes it an offence to perform a restricted act of veterinary science unless a veterinary practitioner.
NT	*Veterinarians Act 1994* *Veterinarians Regulations 1994 [88]*	Castrating and tailing lambs less than 6 months and mulesing sheep are not veterinary services.	Act s 24(1): person should not provide veterinary services unless registered veterinarian. No guidelines under welfare framework for comparison.
QLD	*Veterinary Surgeons Act 1936* *Veterinary Surgeons Regulations 2016 [89]*	Sheep tail docking and castration before 6 months and mulesing less than a year are not acts of veterinary science (Reg 3).	Relatively consistent with the animal welfare legislative framework, which is silent on ages for castration and tail docking but requires that mulesing is performed at less than 12 months. It is an offence to practise veterinary science if not a veterinary surgeon unless other than for fee or reward s25m(1) and (2) of the Act.
SA	*Veterinary Practice Act* *Veterinary Practice Regulations 2017 [90]*	Sheep tail docking and castration less than 3 months and mulesing are not acts of veterinary surgery (Reg 5(2))	Animal Welfare Regulations are silent onage of performance. In animals over 3 months, this is an act of veterinary surgery- performance of which is an offence under s 39 of the Act (but only if done for fee or reward), which likely makes performance a non-issue for farmers.
TAS	*Veterinary Surgeons Act 1987* *Veterinary Surgeons Regulations 2012 [91]*	Tail docking, castration, or mulesing of lambs that are 6 months old or less is not a veterinary service	Codes generally silent on ages, although mulesing may be performed up to 12 months. This conflicts with veterinary regulations, which suggest that farmers may only perform this for up to 6 months.
VIC	*Veterinary Practice Act 1997* (and Regulations) [92]	None identified	N/A
WA	*Veterinary Practice Act 2021* and *Veterinary Practice Regulations 2022 [93]*	Tailing or mulesing of lambs or castrating animals that have not reached 12 months, if performed using humane methods, as not an act of veterinary medicine.	Relatively consistent with animal welfare framework. However, these provisions are conservative, extending the upper age limit for procedure performance. 56 (1) of Act makes it an offence to carry out act of veterinary medicine if not a veterinarian, veterinary nurse, or authorised person.

## Data Availability

All data are contained within this paper.

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
