# Peer review of "How Well Does Australian Animal Welfare Policy Reflect Scientific Evidence: A Case Study Approach Based on Lamb Marking"

_animals, 2023, doi:10.3390/ani13081358_

Round 1
Reviewer 1 Report
This is a well written paper on a timely and very important topic. I have several suggestions where I feel the manuscript could be refined but they are merely suggestions to improve the flow. There are a few grammatical changes recommended as well. The authors have done a great job presenting a lot of data in a concise and clear manner.
My suggestions for improvement are as follows:
Lines 43-46- "Legal frameworks for animal welfare protection should provide the scaffolding upon which a consensus can be reached that maximises animal welfare based on scientific evidence and meets socially respected requirements to continue production at a high standard and maintain social licence to operate." This is a long sentence. Could you revise it or break it into 2 sentences? The information is important but the reader gets lost in the volume of information included in a single sentence.
Line 64- commas are needed between "where and in" and "context and a"
Line 67- remove "in contrast with the UK" as that is already implied in the sentence.
Line 68-70- "Several producer and consumer surveys have also highlighted some inconsistencies between consumer and farmer perceptions of these husbandry procedures and the relevant legislation" This sentence is awkward in its placement. Maybe start with "In addition, " or move it earlier in the paragraph.
Line 89- need a comma between "systems and lambs"
Line 93- need a comma between "docking and with" and between "mulesing and"
Line 99- can you provide more information about immunocastration? what is used or how is it performed? Where is a product licensed or where is this procedure allowed?
Line 124-126 "All of these factors will influence the duration and severity of pain experienced by the animals and our current understanding of the degree of influence these different factors have, is limited" This is a long sentence. Could you revise it and split it into 2 sentences? Maybe take out the "and" and create the second sentence starting with "Our"?
Line 128-130- "We will discuss the science around the impact of different methods, performance at different ages, and associated pain mitigation strategies in the following sections" The first part of this sentence is awkward. Could you revise?
Section 4.1 (Tail Docking)- You introduced the abbreviations for the states and territories in section 3.2. In this section, you are using the full names for the states and territories again. Once the abbreviations have been introduced, they can be used throughout the manuscript. Same with AHA and Animal Health Australia (lines 332 and 338). If you will use the abbreviations, use them consistently throughout the manuscript.
Line 479- Need a comma between "states and the"
Line 480- Need a comma between "states and these"
Lines 504-509- "This transfer of skills is highly valued within farming communities and should be respected and considered as recommendations and standards are updated and this knowledge is disseminated but it is important to also consider a science-based approach to guideline formation, and indeed transparency of this approach i.e., it is clear to reviewers what the scientific basis of the recommendations is" This is a long, complex sentence. Please consider revising and splitting into 2-3 sentences.
Line 529- commas should bracket "of course"
Line 563- comma is needed between "life and the"
Line 601- What is MNT? I don't remember that acronym being defined earlier. Apologies if I missed it.
Line 616- Recommend remove "However and start the sentence with From. Add a comma after "approach".
Line 621- Recommend to end the sentence after "appropriately" and start a new sentence with "In"
Line 636- add a comma after "discussion"
Line 641- Recommend to end the sentence after "were" and start a new sentence with "Farmers"
Line 696- What is "jump racing perhaps a topical example"? That may be a regional or local expression that is not understood universally. Please clarify or remove.
Line 708- Please define EU for the first time before using the acronym.
Line 722- please add a comma between "process and if"
Line 723- please add a comma between "environmentally and an"
Lines 729-732- "For example, in Australia (need comma) when generating the new Standards and Guidelines (need comma) a review of the literature was used in development of the Australian Animal Welfare Standards and Guidelines for Pigs (need comma) which whilst funded by the Australian Pork Industry was subject to independent peer review." This is a long, awkward sentence that is hard to follow. Could you revise and split it into 2 sentences?
Line 735- need comma between "jurisdictions and the"
Lines 764-768- Good conclusion on Victoria. Is there a state or territory that contrasts with them to demonstrate your key point on the differing legislation?
Paragraph (lines 759-768) and Paragraph (lines 770-775) do not tie together well. The final conclusion/summary struggles to summarize the key points that were well delineated in the manuscript.
Author Response
Dear Reviewer 1,
Thank you very much for your time and your kind words and suggestions. We have carefully considered your suggestions and responded to all of them below.
Comments and suggestions from reviewer 1:
- Lines 43-46- "Legal frameworks for animal welfare protection should provide the scaffolding upon which a consensus can be reached that maximises animal welfare based on scientific evidence and meets socially respected requirements to continue production at a high standard and maintain social licence to operate." This is a long sentence. Could you revise it or break it into 2 sentences? The information is important but the reader gets lost in the volume of information included in a single sentence.
- Line 46-50 changed to: Legal frameworks for animal welfare protection should provide the scaffolding upon which a consensus can be reached. This consensus should optimise animal welfare based on available scientific evidence and meet socially respected requirements to promote a high standard of production and maintain social licence to operate.
- Line 64- commas are needed between "where and in" and "context and a"
- Commas added
- Line 67- remove "in contrast with the UK" as that is already implied in the sentence.
- Segment deleted
- Line 68-70- "Several producer and consumer surveys have also highlighted some inconsistencies between consumer and farmer perceptions of these husbandry procedures and the relevant legislation" This sentence is awkward in its placement. Maybe start with "In addition, " or move it earlier in the paragraph.
- Sentence changed, but left in place: In addition, several surveys in Australia and internationally, have highlighted inconsistencies between consumer and farmer perceptions of mulesing [6] and other husbandry procedures and the relevant legislation [6-9].
- Line 89- need a comma between "systems and lambs"
- Comma added
- Line 93- need a comma between "docking and with" and between "mulesing and"
- Commas added
- Line 99- can you provide more information about immunocastration? what is used or how is it performed? Where is a product licensed or where is this procedure allowed?
- Lines 117-123 changed to: Immunocastration is a newer alternative to traditional methods of castration that is doesn’t cause pain [12, 13]. This method blocks the normal functioning of the hypothalamic-pituitary-gonadal axis by administering two doses of vaccine against gonadotrophin-releasing hormone (GnRH). Immunisation against GnRH results in suppression of testosterone production and spermatogenesis [12, 14]. While this technique has been widely adopted in pigs, there is currently no licenced product in sheep at this stage [13-15].
- Line 124-126 "All of these factors will influence the duration and severity of pain experienced by the animals and our current understanding of the degree of influence these different factors have, is limited" This is a long sentence. Could you revise it and split it into 2 sentences? Maybe take out the "and" and create the second sentence starting with "Our"?
- Changed to: All of these factors will influence the duration and severity of pain experienced by the animal. Our current understanding of the degree of influence these factors have pain experience is limited.
- Line 128-130- "We will discuss the science around the impact of different methods, performance at different ages, and associated pain mitigation strategies in the following sections" The first part of this sentence is awkward. Could you revise?
- Changed to: In the following sections we will discuss the scientific evidence on the impact of different methods of performing common husbandry procedures, the ages at which they’re performed, and associated pain mitigation strategies.
- Section 4.1 (Tail Docking)- You introduced the abbreviations for the states and territories in section 3.2. In this section, you are using the full names for the states and territories again. Once the abbreviations have been introduced, they can be used throughout the manuscript. Same with AHA and Animal Health Australia (lines 332 and 338). If you will use the abbreviations, use them consistently throughout the manuscript.
- Changed abbreviations as recommended
- Line 479- Need a comma between "states and the"
- Commas added
- Line 480- Need a comma between "states and these"
- Commas added
- Lines 504-509- "This transfer of skills is highly valued within farming communities and should be respected and considered as recommendations and standards are updated and this knowledge is disseminated but it is important to also consider a science-based approach to guideline formation, and indeed transparency of this approach i.e., it is clear to reviewers what the scientific basis of the recommendations is" This is a long, complex sentence. Please consider revising and splitting into 2-3 sentences.
- Lines 580-584 changed to: “This transfer of skills is highly valued within farming communities and should be respected and considered as recommendations and standards are updated. It is important to also consider a science-based approach to guideline formation, and indeed transparency of this approach i.e., it is clear to reviewers what the scientific basis of the recommendations is.”
- Line 529- commas should bracket "of course"
- Commas added
- Line 563- comma is needed between "life and the"
- Commas added
- Line 601- What is MNT? I don't remember that acronym being defined earlier. Apologies if I missed it.
- Abbreviation changed to mechanical nociceptive threshold
- Line 616- Recommend remove "However and start the sentence with From. Add a comma after "approach".
- Changed as recommended
- Line 621- Recommend to end the sentence after "appropriately" and start a new sentence with "In"
- Changed as recommended
- Line 636- add a comma after "discussion"
- Changed as recommended
- Line 641- Recommend to end the sentence after "were" and start a new sentence with "Farmers"
- Changed as recommended
- Line 696- What is "jump racing perhaps a topical example"? That may be a regional or local expression that is not understood universally. Please clarify or remove.
- Removed example
- Line 708- Please define EU for the first time before using the acronym.
- Changed as recommended
- Line 722- please add a comma between "process and if"
- Changed as recommended
- Line 723- please add a comma between "environmentally and an"
- Changed as recommended
- Lines 729-732- "For example, in Australia (need comma) when generating the new Standards and Guidelines (need comma) a review of the literature was used in development of the Australian Animal Welfare Standards and Guidelines for Pigs (need comma) which whilst funded by the Australian Pork Industry was subject to independent peer review." This is a long, awkward sentence that is hard to follow. Could you revise and split it into 2 sentences?
- Changed to: “For example, when the new Standards and Guidelines were generated in Australia, a review of the literature was used in development of the Australian Animal Welfare Standards and Guidelines for Pigs. Whilst this review was funded by the Australian Pork Industry, it was subject to independent peer review [24].”
- Line 735- need comma between "jurisdictions and the"
- Changed as recommended
- Lines 764-768- Good conclusion on Victoria. Is there a state or territory that contrasts with them to demonstrate your key point on the differing legislation?
- Lines 904-908 changed to: “Moreover, enforceability of the Code varies considerably across the states. Vic provides an excellent example of this with a Code that arguably is the most welfare friendly by requiring all mulesed animals to have pain relief and for operators to be formally trained in the procedure. However, this code is a voluntary code of practice in this state and therefore there are no direct ramifications for failing to adhere to these provisions. In comparison, NSW, having adopted the AHA Sheep Welfare Standards and Guidelines, requires that pain relief be used in sheep that are mulesed when they are between 6 to 12 months of age. While these standards are mandatory and violation of them may be used as evidence of an offence in court [126].”
- Paragraph (lines 759-768) and Paragraph (lines 770-775) do not tie together well. The final conclusion/summary struggles to summarize the key points that were well delineated in the manuscript.
- Conclusion changed to:
There is broad consistency across the Australian jurisdictions in relation to specific provisions around procedures at lamb marking, for example recommended age ranges at performance of procedure and the use of anaesthesia in animals over 6 months old. Recommendations for the use of pain relief in animals less than 6 months are non-specific or absent in most states. The scientific evidence surrounding marking procedures indicates that castration, tail docking, and mulesing cause pain acutely and for at least two days post procedure regardless of age. This pain can be mitigated to some extent by NSAIDs and local anaesthetics that are licensed for use in sheep in Australia. There is a clear disconnect between the relevant legislation and scientific evidence. Legislation is lacking in traceability back to the science through no direct referencing. It also remains unclear from documents in the public domain what the scientific basis was, and the process for assimilating the science on sheep husbandry during the creation of the new sheep standards and guidelines.
Yours sincerely,
Charlotte Johnston
Reviewer 2 Report
This article is both novel and significant in that it addresses the neglected area of incorporation of scientific knowledge into animal welfare laws. The approach the authors take to investigating whether and how science has been incorporated is clear and appropriate. The authors are to be commended for their initiative in researching this important area.
I recommend the article be published with minor amendments.
My most significant query relates to the reflection of the findings in the conclusion. The conclusion states that '[t]hese provisions also seem to broadly match with the scientific evidence available'. This is not the impression I received when reading the article. For example, on p16 reference is made to science showing the negative impact of painful procedures early in life and then questioning why the legislation does not require both early age and and pain mitigation strategies. This would suggest lack of implementation of science in legislation (other reasons are suggested for the disconnect, but not scientific ones).
Other minor queries:
- The opening sentence states that animal farming was once widely accepted but is not any more. This glosses over the significant changes in farming methods over time, e.g. increase in 'factory farm' like conditions.
- Reference is made to the UK FTA. Note a similar provision is anticipated in the forthcoming EU FTA.
- On p3, line 141, should read 'state and territory acts'
- Circa p 4-6, ensure consistent use of farm animals versus farmed animals. My preference would be farmed animals.
- Mulesing occurs only in Australia but is banned elsewhere. It is not clear to me why this is and which position is better supported by scientific research (perhaps I am missing something?)
- Page 16 at 641-2. This broadbrush statement appears to be supported by the single piece of evidence. Can more be provided or referenced?
- Discussion of 'Farmer perception' pp16-17 indicates a clear need to better educate farmers re legislative requirements (rather than just pursue change through 'other methods') - this should be made clear.
Well done!
Author Response
Dear Reviewer 2,
Thank you very much for your time and kind words and suggestions. We have carefully considered your suggestions and made the following changes detailed below.
Comments and suggestions from reviewer 2:
- My most significant query relates to the reflection of the findings in the conclusion. The conclusion states that '[t]hese provisions also seem to broadly match with the scientific evidence available'. This is not the impression I received when reading the article. For example, on p16 reference is made to science showing the negative impact of painful procedures early in life and then questioning why the legislation does not require both early age and and pain mitigation strategies. This would suggest lack of implementation of science in legislation (other reasons are suggested for the disconnect, but not scientific ones).
- Conclusion revised as above (Lines 881-894)
- The opening sentence states that animal farming was once widely accepted but is not any more. This glosses over the significant changes in farming methods over time, e.g. increase in 'factory farm' like conditions.
- Lines 36-39 changed to: The farming of animals, once widely accepted by society, is now under growing scrutiny as animal production becomes more intensive and social attitudes toward the use of animals changes. This scrutiny comes from numerous groups of differing social, scientific, and political backgrounds.
- Reference is made to the UK FTA. Note a similar provision is anticipated in the forthcoming EU FTA.
- Lines 72-74 added to include the EU FTA: A similar concern may arise alongside trade with one of the world’s biggest trading blocs, the European Union (EU), in the forthcoming trade agreement [10].
- On p3, line 141, should read 'state and territory acts' with
- Changed as recommended
- Circa p 4-6, ensure consistent use of farm animals versus farmed animals. My preference would be farmed animals.
- Changed to farm animals – farm animals is used more consistently throughout the literature and legislation
- Mulesing occurs only in Australia but is banned elsewhere. It is not clear to me why this is and which position is better supported by scientific research (perhaps I am missing something?)
- Thank you for this comment. I have included further information about the efforts to phase out mulesing in Australia with reference to the phasing out of mulesing in New Zealand.
- Lines 462-481: The procedure has received global scrutiny for its negative impacts on welfare and is banned in most countries, with our close neighbours New Zealand banning the procedure in 2018 [71]. Phasing out mulesing in New Zealand was largely industry-led and took roughly 5 years [72]. Differences in the Australian climate, wool industry, predominance of the Merino, and larger enterprises compared to the New Zealand wool industry have substantially delayed the phasing out of mulesing in Australia [72]. Nevertheless, Australian livestock industries are working towards phasing out mulesing through research into alternative ways of preventing flystrike, including breeding programmes to reduce wrinkle scores [21, 68], and developing extension strategies to educate and support producers transitioning to non-mulesing operations [72]. Flystrike remains a major concern and the risk of flystrike is expected to increase with the emergence of chemical resistance limiting the efficacy of chemicals used for prevention and treatment [68, 69]. It has also been suggested that the distribution and abundance of the fly population may increase with climate change [68], thus increasing the risk of flystrike. This highlights the importance of continued support for research investigating flystrike prevention and treatment to foster a sustainable sheep and wool industry that is able to meet consumer demands and maintain the social license to operate [72].
- Page 16 at 641-2. This broadbrush statement appears to be supported by the single piece of evidence. Can more be provided or referenced?
- I chose to add further detail to the inappropriate use of analgesics to emphasise the importance of farmer education and the factors that may influence dissemination of relevant knowledge.
- Lines 693-731 changed to: From a practical approach, the current analgesic options available to farmers are somewhat limited as they have a relatively short duration of action (30minutes to 72 hours [79, 109, 110]), peri-operative analgesia or anaesthesia requires prior administration and double-handling, they can also be impractical or difficult to administer during marking, and they can be cost prohibitive. There is also evidence that education on the use of pain relief is lacking as a number of surveys show that producers are not always using available analgesics appropriately [109, 111]. For example, a survey of Australian sheep producers found that, of the 30% of producers using pain relief for rubber ring castration, over half (58%) reported using Tri-SolfenÒ which is an unsuitable analgesic for this method and indicates a misunderstanding of the mode of action of this product [109]. Additionally, the use of suitable multimodal analgesia (NSAID and appropriate local anaesthetic), which is current best practice [79, 80], was used by less than 10% of producers for castration and tail docking (1% and 7.7%, respectively) [109]. Education programmes covering the recognition of pain in animals, the detrimental effects of pain, and the use of pain relief are clearly an important part of promoting the appropriate use of pain relief for husbandry procedures [8, 43, 109]. Effective dissemination of new scientific findings and legislative changes or recommendations is crucial for constructive development within the farming sector. Government departments and agencies are not always seen as trusted sources of information [43, 112]. Whereas, experienced farmers are seen as trusted advisors within farming communities [43] and knowledge and experience are often passed down between farmers and within families through informal training [43, 44]. These factors should be taken into consideration when developing intervention and education strategies to effect real change.
- Discussion of 'Farmer perception' pp16-17 indicates a clear need to better educate farmers re legislative requirements (rather than just pursue change through 'other methods') - this should be made clear.
- I chose to add further detail to the inappropriate use of analgesics to emphasise the importance of farmer education and the factors that may influence dissemination of relevant knowledge.
- Thank you for this comment. I have included further information about the efforts to phase out mulesing in Australia with reference to the phasing out of mulesing in New Zealand.
Lines 749-759 changed to: Similarly, Woodruff et al. [42] found that a lack of awareness of the recommended tail docking length was a major factor driving docking practices in 57% of surveyed Victorian sheep farmers that docked tails shorter than three palpable joints. Knowledge and implementation of current legislation and guidance on farming practices among the farming community appears to be a major barrier globally [7, 9, 42, 43]. This leads us to query the efficacy of enforced legislative changes over other methods of knowledge dissemination and changing practice, such as education programs for farmers and other stakeholders. Ultimately, a combination of legislative change and stakeholder-led education programmes is more likely to create sustained and widespread improvements in animal husbandry.
Yours sincerely,
Charlotte Johnston